:ᴑ: PLOS | ONE

# High expression of olfactomedin-4 is correlated with chemoresistance and poor prognosis in pancreatic cancer

Ryotaro Ohkuma[1,2,3], Erica Yada[4], Shumpei Ishikawa[5], Daisuke Komura[5], Hidenobu Ishizaki[6], Koji Tamada[7], Yutaro Kubota[2], Kazuyuki Hamada[2], Hiroo Ishida[2], Yuya Hirasawa[2,8], Hirotsugu Ariizumi[2], Etsuko Satoh[2], Midori Shida[1,9], Makoto Watanabe[1,9], Rie Onoue[1,9], Kiyohiro Ando[1,9], Junji Tsurutani[2,10], Kiyoshi Yoshimura[2,8,9], Takehiko Yokobori[11], Tetsuro Sasada[4], Takeshi Aoki[12], Masahiko Murakami[12], Tomoko Norose[13], Nobuyuki Ohike[13], Masafumi Takimoto[13], Masahiko Izumizaki[3], Shinichi Kobayashi[9], Takuya Tsunoda[2], Satoshi Wada[1,2,9]*

1 Department of Clinical Diagnostic Oncology, Clinical Research Institute for Clinical Pharmacology & Therapeutics, Showa University, Tokyo, Japan, 2 Department of Medicine, Division of Medical Oncology, School of Medicine, Showa University, Tokyo, Japan, 3 Department of Physiology, Graduate School of Medicine, Showa University, Tokyo, Japan, 4 Kanagawa Cancer Center Research Institute, Kanagawa, Japan, 5 Department of Molecular Preventive Medicine, Graduate School of Medicine, The University of Tokyo, Tokyo, Japan, 6 Noile-Immune Biotech, Inc., Tokyo, Japan, 7 Department of Immunology, Graduate School of Medicine, Yamaguchi University, Yamaguchi, Japan, 8 Department of Clinical Immuno Oncology, Clinical Research Institute for Clinical Pharmacology & Therapeutics, Showa University, Tokyo, Japan, 9 Clinical Research Institute for Clinical Pharmacology & Therapeutics, Showa University, Tokyo, Japan, 10 Advanced Cancer Translational Research Institute, Showa University, Tokyo, Japan, 11 Department of Innovative Immune-Oncology Therapeutics, Graduate School of Medicine, Gunma University, Gunma, Japan, 12 Department of Surgery, Division of General and Gastroenterological Surgery, School of Medicine, Showa University, Tokyo, Japan, 13 Department of Pathology and Laboratory Medicine, School of Medicine, Showa University, Tokyo, Japan

* st-wada@med.showa-u.ac.jp

## Abstract

Pancreatic cancer has an extremely poor prognosis, and identification of novel predictors of therapeutic efficacy and prognosis is urgently needed. Chemoresistance-related molecules are correlated with poor prognosis and may be effective targets for cancer treatment. Here, we aimed to identify novel molecules correlated with chemoresistance and poor prognosis in pancreatic cancer. We established 10 patient-derived xenograft (PDX) lines from patients with pancreatic cancer and performed next-generation sequencing (NGS) of tumor tissues from PDXs after treatment with standard drugs. We established a gene-transferred tumor cell line to express chemoresistance-related molecules and analyzed the chemoresistance of the established cell line against standard drugs. Finally, we performed immunohistochemical (IHC) analysis of chemoresistance-related molecules using 80 pancreatic cancer tissues. From NGS analysis, we identified olfactomedin-4 (OLFM4) as having high expression in the PDX group treated with anticancer drugs. In IHC analysis, OLFM4 expression was also high in PDXs administered anticancer drugs compared with that in untreated PDXs. Chemoresistance was observed by *in vitro* analysis of tumor cell lines with forced expression of OLFM4. In an assessment of tissue specimens from 80 patients with pancreatic cancer, Kaplan-Meier analysis showed that patients in the low OLFM4 expression group had a better survival rate than patients in the high OLFM4 expression group. Additionally, multivariate analysis showed that high expression of OLFM4 was an independent prognostic factor

(contact via Showa university) for researchers who meet the criteria for access to confidential data. Ethics Committee of the Showa University School of Medicine Address; 1-5-8, Hatanodai, Shinagawa-ku, Tokyo, 142-8555, Japan Phone; +81-3-3784-8129 Enail; m-rinri@ofc.showa-u.ac.jp

**Funding:** This study was funded by Noile-Immune Biotech, Inc. All funding was performed as the collaborative research project and was only used to purchase the reagents and equipment needed for the experiment. The funder had no additional role in the study design, data collection and analysis, decision to publish, or preparation of the manuscript. The funder also provided support in the form of salaries to HI. The specific roles of these authors are articulated in the 'author contributions' section support in the form of salaries to HI. The specific roles of these authors are articulated in the 'author contributions' section.

**Competing interests:** The authors have read the journal's policy and the authors of this paper have the following competing interests: HI is paid employees of Noile-Immune Biotech, Inc., clinical stage biotechnology company. There are no patents, products in development or marketed products associated with this research to declare. This does not alter our adherence to PLOS ONE policies on sharing data and materials.

predicting poor outcomes. Overall, our study revealed that high expression of OLFM4 was involved in chemoresistance and was an independent prognostic factor in pancreatic cancer. OLFM4 may be a candidate therapeutic target in pancreatic cancer.

## Introduction

Pancreatic cancer is the most aggressive human malignancy and the fourth leading cause of cancer-related death in the United States of America (USA) [1] and Japan [2]. Moreover, pancreatic cancer causes more than 200,000 deaths worldwide every year and is associated with an overall 5-year survival rate of less than 6% after diagnosis in the USA [1,3,4].

Overall survival rates for patients with pancreatic cancer have not improved significantly in the past 30 years, and the mortality rate is similar to the incidence owing to the late diagnosis in most patients. Thus, only approximately 20% of tumors are resectable at presentation [4], and development of improved methods for early diagnosis is urgently needed. Another reason for the high mortality rates is resistance to chemotherapy and radiotherapy [5,6]. Despite our broader understanding of pancreatic cancer biology, gemcitabine (GEM), which was approved for pancreatic cancer treatment approximately 20 years ago and fundamentally changed cancer treatment, remains the standard treatment for this aggressive cancer [7–9]. In addition, no studies have supported the appropriate regimen for second-line chemotherapy. Thus, novel therapeutic strategies for the treatment of pancreatic cancer are required. As a key drug used for the treatment of pancreatic cancer, GEM treatment can often lead to chemoresistance. Thus, in order to improve the prognosis of patients with pancreatic cancer, novel approaches are needed to overcome chemoresistance to GEM.

Many types of cancer cell lines have been used in research studies. However, because these cell lines are cultured under artificial conditions, they do not necessarily reflect the actual kinetics and phenotypes of cancer cells. Animal models are often used in preclinical studies for predicting the efficacy and possible toxicities of anticancer drugs in patients with cancer [10,11]. PDX models have attracted attention in recent years for assessment of the efficacy of anticancer drugs [12,13] and for biomarker development and testing. Additionally, these models have been used to clarify the microenvironment and characteristics of cancer cells. In PDX models, cancer cells or small tumor tissues derived from patients are injected into immune-deficient mice and retain similar morphology, architecture, and molecular signatures as the original cancers; thus, these models could have applications in rapid screening of potential therapeutics. PDX models could preserve clinical information from the donor patient, enabling accelerated cancer research by simulating the human cancer microenvironment [14,15]. Therefore, we established PDXs for use in this this study.

In this study, we aimed to identify novel chemoresistance-related molecules in pancreatic cancer using pancreatic cancer PDXs. We then analyzed the role of olfactomedin-4 (OLFM4), which was identified as a chemoresistance-related protein, in chemoresistance in an *in vitro* model and evaluated the expression and prognostic ability of OLFM4 by immunohistochemical (IHC) analysis in 80 pancreatic cancer tissues from human patients.

## Materials and methods

### Establishment of pancreatic cancer PDXs

Immune-deficient NSG mice were obtained from Jackson Laboratories (Sacramento, CA, USA). All animals were housed in plastic cages in a pathogen-free state, at a temperature of

22 ± 1˚C with 45% ± 10% humidity and a 12-h light/12-h dark cycle. All animals were fed a standard diet and allowed free access to water. All experiments involving laboratory animals were performed in accordance with the care and use guidelines of the Kanagawa Cancer Center Research Institute. The study was approved by the Research Ethics Committee of Kanagawa Cancer Center Research Institute (approval no. 176).

Tumor tissues from surgical specimens removed from patients with pancreatic cancer were transplanted subcutaneously into 6–12-week-old NSG mice using transplantation needles (S1A Fig) [16,17]. The details of patients with pancreatic cancer whose tumor tissues were used to generate the PDX mouse models are shown in S1 Table. All mice received the anesthesia with inhaled isoflurane, and euthanasia was performed by cervical dislocation before tumor tissues were excised. The PDXs prepared by transplantation were designated as Generation 1 (G1). When the tumor volumes in G1 mice reached 1,000 mm$^3$, the tumor tissues were removed and re-implanted into other NSG mice (S1B and S1C Fig). We repeated the passaging and succeeded in generating PDXs up to G7. In this study, 10 lines of PDXs were established and used. The process for removing surgical specimens of tumor tissues from patients with pancreatic cancer and transplanting them into mice was also approved by the Research Ethics Committee of Kanagawa Cancer Center Research Institute (approval no. 176).

### Characteristics of pancreatic cancer PDXs

We conducted IHC staining and gene analyses to verify whether PDXs retained the characteristics of patient tissues, even after repeated passaging. Tumor tissues were fixed with 10% formalin and embedded in paraffin. Tissues were then sliced into 4-μm-thick sections and subjected to standard hematoxylin and eosin (HE) staining or IHC analysis. After deparaffinization and washing in phosphate-buffered saline, endogenous peroxidase activity was inactivated by treatment with 0.3% $H_2O_2$ for 30 min. Primary antibodies specific for human leukocyte antigen (HLA) class I -A, B, C (dilution 1:500; Hokudo, Sapporo, Japan) were applied and incubated for 60 min to demonstrate that tumor cells in PDXs were derived from patient tissues. Simple stain MAX-PO (M) (Nichirei Biosciences, Japan) was used as a secondary antibody and incubated with the samples for 1 h. After washing, cells were visualized by 3,3′-diaminobenzidine (DAB) staining and $H_2O_2$ treatment.

In order to investigate the characteristics of gene mutations, we conducted DNA/RNA extraction and high-throughput sequencing. The tumor tissues were cut into small pieces and frozen in liquid nitrogen as soon as possible after tumor-bearing mice were sacrificed. The frozen tissues were crushed using a Cryo-Press (Microtec, Funabashi, Chiba, Japan). Total DNA/RNA from crushed tissues was isolated and purified with a ZR-Duet DNA/RNA MiniPrep kit (ZYMO RESEARCH, Irvine, CA, USA). DNA/RNA quality was then assessed using an Agilent 2100 Bioanalyzer (Agilent Technologies, Santa Clara, CA, USA). cDNA libraries were constructed using these RNAs, and libraries were subsequently sequenced on an Illumina HiSeq2000 platform at BGI (Shenzhen, China). Approximately 5 Gb of raw reads was generated for each of the samples.

### Antitumor effects in the PDX model

When tumor volumes reached 200–400 mm$^3$ in the pancreatic cancer PDX, the PDX was administered anticancer drugs (gemcitabine [GEM] monotherapy, GEM and nab-paclitaxel [nab-PTX] combination therapy) or saline as an untreated control. Tumor volume (mm$^3$) was calculated using the following formula: (smaller measurement [mm]$^2$ × larger measurement [mm]) / 2. Anticancer drugs or saline was intraperitoneally administered as shown in S2 Fig. The tumor volumes in both groups were measured every week.

## Identification of chemoresistant molecules

Tumor tissues were collected from PDXs treated with chemotherapy or saline when the tumor volume of the PDX in the control group exceeded 1,500 mm$^3$. Then, DNA/RNA was extracted from these tissues, and next-generation sequencing (NGS) was performed using an HiSeq 4000 system with paired-end sequencing. The read lengths from whole genome sequencing were 2 × 100 bp for GEM monotherapy and 2 × 150 bp for GEM and nab-PTX combination therapy.

## Transcriptome analysis

Paired-end reads were mapped to all RefSeq transcripts of humans (hg38 coordinates) and mice (mm10 coordinates) using bowtie 1.1.2 [18], allowing up to one mismatch, and reads mapped to both species or to multiple genes were discarded. To avoid bias derived from read length differences, only the first 100 bp of each read was used for mapping for samples with read lengths of 150 bp. After reads mapped to noncoding transcripts were removed, the remaining reads were used to estimate the gene expression profiles of human cancer cells and mouse stromal cells according to the methods described in our previous report [19]. Gene expression values were normalized for cancer cells and stromal cells independently, such that the sum of the expression values below the 95th percentile was 300,000.

## Bioinformatics analysis using The Cancer Genome Atlas (TCGA) database

TCGA is a National Cancer Institute (NCI) effort to profile at least 20 different tumor types using genomic platforms and to generate raw and processed data and make these data available to all researchers. TCGA has released large amounts of RNA sequencing data from patients with pancreatic adenocarcinoma. TCGA RNA Seq data for pancreatic cancer were down-loaded and used to generate Kaplan-Meier curves according to OLFM4 expression.

## IHC analysis in the PDX model

IHC staining of OLFM4 was performed after treatment with anticancer drugs in PDX mice. IHC staining for OLFM4 was carried out as described below. Because the localization of the tumor cells varied within the pathological specimens, the tumor portion in the slide was photographed at 200× magnification in three places. The number of pixels for each field was calculated using Image J and digitized [20].

## Cell viability assay

HeLa cells were purchased from American Type Culture Collection (Manassas, VA, USA). PANC-1, KP2, MIA-PaCa-2, and SUIT-2 were obtained from Kanagawa Cancer Center. HeLa cells were grown in DMEM (Thermo Fisher Scientific, San Jose, CA, USA) containing 5% heat-inactivated fetal bovine serum (FBS, GE Healthcare, Uppsala, Sweden), 100 U/mL penicillin, 100 μg/ml streptomycin, 2 mM L-glutamine, and non-essential amino acids. MIA PaCa-2 cells were grown in DMEM (Thermo Fisher Scientific, San Jose, CA, USA), and PANC-1, KP2, and SUIT-2 cells were grown in RPMI (Thermo Fisher Scientific, San Jose, CA, USA). Both media were supplemented with 10% heat-inactivated fetal bovine serum (FBS, GE Healthcare, Uppsala, Sweden), 100 U/ml penicillin, and 100 μg/ml streptomycin. Cell lines were incubated at 37°C in a humidified atmosphere containing 5% CO2.

Each cell line was seeded in 6-well plates for 24 h before transfection. After 24 h of growth, transfection was carried out using following transfection regents: FuGENE 6 (Promega, Madison, WI, USA) for HeLa cells; Lipofectamine 3000 Reagent (Thermo Fisher Scientific

Inc., Waltham, MA, USA) for MIA-Paca-2 and PANC-1 cells; and Lipofectamine RNAiMAX Transfection Reagent (Thermo Fisher Scientific Inc., Waltham, MA, USA) for SUIT-2 and KP2 cells according to the manufacturer's protocol. OLFM4 expression plasmid (pCMV6-Ac-OLFM4-GFP tag plasmids, NM_006418) was purchased from OriGene Technologies, Inc. (Rockville, MD, USA) and an empty vector was used as a control. OLFM4 specific siRNA (OLFM4 Silencer Select siRNA, Cat# 4392420) and negative control siRNA (Silencer Select siRNA Negative control #1 siRNA, Cat# 4390843) were purchased from Thermo Fisher Scientific Inc. (Waltham, MA, USA). The specificity and efficacy of siRNA were initially evaluated.

Twenty-four hours after transfection, cells were seeded at $1 \times 10^4$ cells/well, and GEM was added at various concentrations (0–100 nM) 24 h later. Seventy-two or 48 h after incubation, cell viability was measured by MTT Reagent or alamarBlue Cell Viability Regent (Thermo Fisher Scientific Inc., Waltham, MA, USA), according to manufacturer's protocol. The cell number was determined, with that in untreated cells set as 100%, and the rate of change was calculated.

## Immunohistological analysis of samples from patients with pancreatic cancer

The study comprised a consecutive cohort of 80 patients with pancreatic cancer who had undergone surgical resection at Showa University Hospital in Tokyo, Japan from January 1, 2008 to December 31, 2017. Patients with pancreatic cancer were histopathologically diagnosed between January 1, 2008 and December 31, 2017 at the same hospital. Patients with no previous history of chemotherapy (GEM monotherapy, GEM and nab-PTX combination therapy) were included. This study protocol was approved by the Ethics Committee of the Showa University School of Medicine (Tokyo, Japan; approval no. 2611), and all study procedures adhered to the principles of the Declaration of Helsinki. Consent was not obtained from individual patients. However, patients were notified of the details of the study and were given the right to refuse study participation. Additionally, our study was approved by the Ethics Committee of the Showa University School of Medicine (approval no. 2611) in accordance with Japanese ethical guidelines. All data were fully anonymized before being accessed.

We used archived formalin-fixed, paraffin-embedded (FFPE) tumor tissues from resected specimens from patients who were histopathologically diagnosed for immunohistological analysis. Histological classifications were based on the World Health Organization system. Tumor staging was performed according to the criteria described in the UICC TNM classification (7[th] edition, 2009). The patients included in this study were diagnosed with stage IIA or stage IIB disease according to the 7[th] edition of UICC.

Immunohistological analysis was performed to evaluate OLFM4 protein expression in human pancreatic cancer tissues. FFPE tissue sections (3 μm thick) were analyzed using a Leica Bond system with standard protocols, as follows. Briefly, sections were pretreated using heat-mediated antigen retrieval with sodium citrate buffer (pH 6 for anti-OLFM4 antibodies) for 20 min. Sections were incubated with primary antibodies against OLFM4 (1:1000 dilution; cat. no. ab85046; Abcam, Cambridge, UK) for 15 min at room temperature and detected using a horse radish peroxidase-conjugated compact polymer system. DAB was used as the chromogen. Sections were then counterstained with hematoxylin. Finally, sections were viewed under a bright-field microscope.

The stained tissue sections were reviewed and scored separately by two pathologists who were blinded to the clinical parameters. The degree of immunostaining was based on the intensity of staining and percentage of cells stained. Staining intensity was graded according to the following criteria: 0, negative; 1, weak; 2, moderate; and 3, strong (uncolored, light yellow,

yellowish brown, and brown, respectively). Staining percentages were graded according to the proportion of positively stained tumor cells, as follows: 0 for less than 5% positive tumor cells; 1 for 5–9% positive tumor cells; 2 for 10–29% positive tumor cells; and 3 for greater than or equal to 30% positive tumor cells. Using this method of assessment, we evaluated OLFM4 expression based on the staining index (scored as 0, 1, 2, 3, 4, 6, or 9). An optimal cut-off value was identified as follows: a staining index score of less than 2 was used to indicate low OLFM4 expression, and an index score of more than 3 was used to define tumors with high OLFM4 expression.

For analysis of clinicopathological factors, invasive factor was evaluated based on the classification of pancreatic cancer recommended by the Japan Pancreas Society for further stratification. Lymphatic invasion was graded according to the following criteria: ly0, negative; ly1, weak; ly2, moderate; and ly3, strong, according to the degree of invasion. Venous invasion was graded as v0, v1, v2, and v3 according to the same descriptions.

## Statistical analysis

All data were analyzed for statistical significance using JMP Pro 14.0 software (SAS Institute Inc., Cary, NC, USA). Associations between the IHC status of OLFM4 expression and various clinicopathological characteristics were evaluated using Student's t tests or Pearson chi-square tests. Classification and regression tree analysis was used to assess the optimal prognostic cut-off for OLFM4 expression in overall survival (OS). Kaplan Meier analysis and the log rank tests were applied to estimate differences in OS according to high and low OLFM4 expression. OS was defined as the interval in months between the initial pancreatic resection surgery and either death or the last observation. Univariate and multivariate analyses were based on the Cox proportional hazards regression model. All tests were two-sided, and results with $p$ values of less than 0.05 were considered statistically significant.

## Results

### Establishment of a pancreatic cancer PDX model

We have generated many PDX models using patient tumor tissue to date [17]. In this experiment, we successfully established 10 PDX lines for pancreatic cancer (Table 1). In order to verify whether these PDXs retained the characteristics of patient-derived tumors, histopathological analysis and gene analysis were performed using the original tumor tissues from patients and PDXs. Initially, the tumor tissues were stained with HE and analyzed by two pathologists. A comparison of the histopathological findings revealed that the histological

**Table 1. Pancreatic cancer PDXs established in this study.**

| PDX No. | Primary/Metastatic | Pathogenic diagnosis | Generation of PDX |
|---|---|---|---|
| #1 | Primary | Moderately differentiated tubular adenocarcinoma | G4 |
| #2 | Lymph node | Poorly differentiated tubular adenocarcinoma | G5 |
| #3 | Primary | Moderately differentiated tubular adenocarcinoma | G5 |
| #4 | Primary | Well-differentiated tubular adenocarcinoma | G6 |
| #5 | Primary | Poorly differentiated tubular adenocarcinoma | G6 |
| #6 | Primary | Adenosquamous carcinoma | G5 |
| #7 | Primary | Moderately differentiated tubular adenocarcinoma | G6 |
| #8 | Primary | Moderately differentiated tubular adenocarcinoma | G7 |
| #9 | Primary | Well-differentiated tubular adenocarcinoma | G6 |
| #10 | Primary | Poorly differentiated tubular adenocarcinoma | G6 |

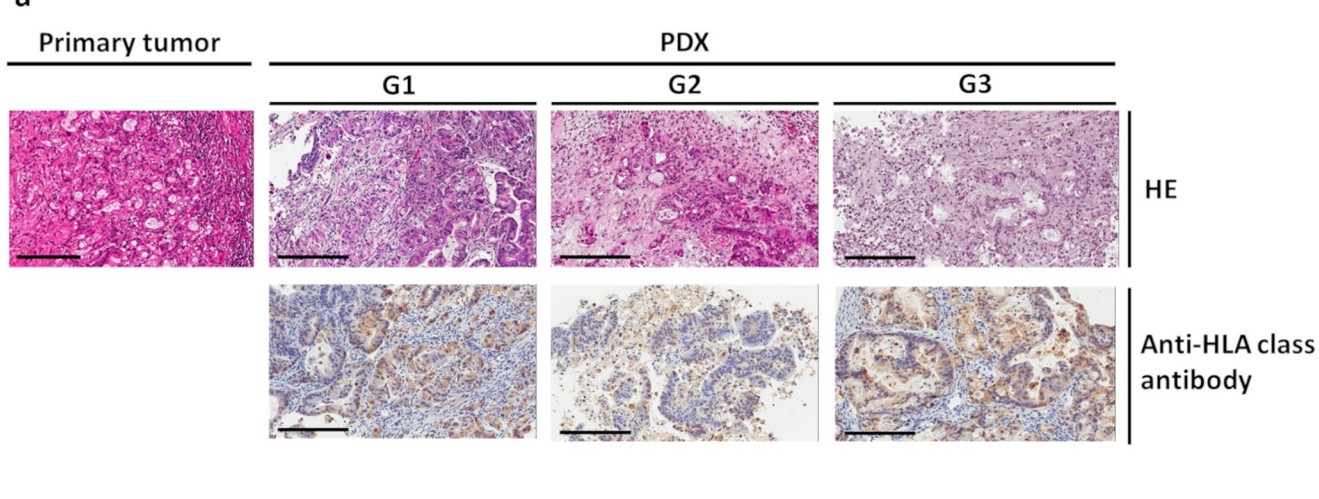

**Fig 1. Establishment of pancreatic cancer PDXs.** (a) Preserved morphological characteristics observed in xenograft tumors in NSG mice. HE staining and immunohistochemistry for anti-HLA class I antibodies are shown for both the primary tumor and each generation of PDXs (bar: 200 μm). The pathological diagnosis of the primary tumor was poorly differentiated tubular adenocarcinoma. Patient-deriver cancer cells (HLA+ cells) were preserved after passaging, and the morphological characteristics were maintained in the xenograft. (b) Preserved genetic alterations in the xenograft tumors in NSG mice. Mutations were detected by next-generation sequencing. The gene mutations found in the patient's tumor cells were consistent with the gene mutations found in the PDX model prepared from the patient.

features of patient specimens were retained, even when PDX passages were repeated (Fig 1A). HE staining was performed in all cases, then in S3 Fig we presented the histological features of nine cases other than the case of Fig 1A. Furthermore, immunohistological analysis showed that the expression of the HLA class I molecule on the tumor cells in PDX specimens was also retained, even after PDX passaging. HLA class I expression supported that tumor cells from PDXs were derived from patient tissues in terms of histological characteristics. In an analysis of gene mutations, we found that mutations in patient samples were maintained in corresponding PDXs (Fig 1B). Thus, the PDXs used in this study retained the pathological features and genetic characteristics of the original pancreatic cancer tissues from patients.

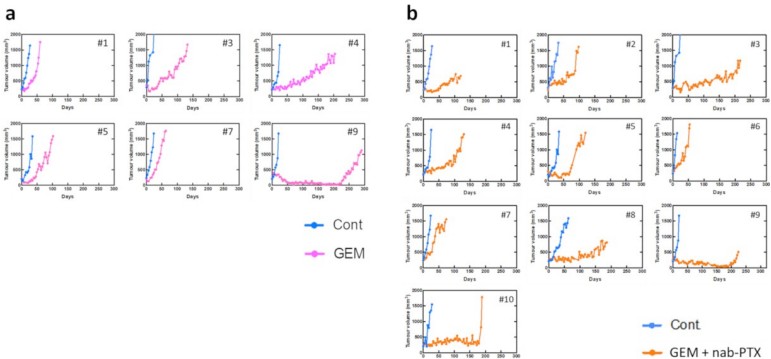

**Fig 2. Tumor growth curves after chemotherapy.** (a) PDX mice were treated with GEM (n = 6) or (b) GEM + nab-PTX (n = 10) after the tumor volume became more than 1500 mm$^3$.

### Identification of a chemoresistance-related molecule

To identify chemoresistant molecules, the antitumor effects of standard treatments for pancreatic cancer were examined using our PDX model. PDXs were treated with GEM monotherapy or GEM plus nab-PTX combination therapy, and tumor growth was examined (Fig 2). The antitumor effects were transiently observed in some cases in the treatment group, but tumor growth was observed in all PDXs. Tumor tissues were collected from PDXs treated with anti-cancer drugs or saline when the tumor volume in the control group exceeded 1,500 mm$^3$. The pathological findings for the collected samples are shown in Fig 3. Although the number of tumor cells was decreased by chemotherapy, the cells recovered over time.

Next, RNA was extracted from tumor tissues, and mRNA expression was analyzed using NGS (Fig 4A). Using the data for normalized expression (NE) values, the ratio between the control and treated PDXs was calculated. GEM single-agent therapy (6 lines) and GEM plus

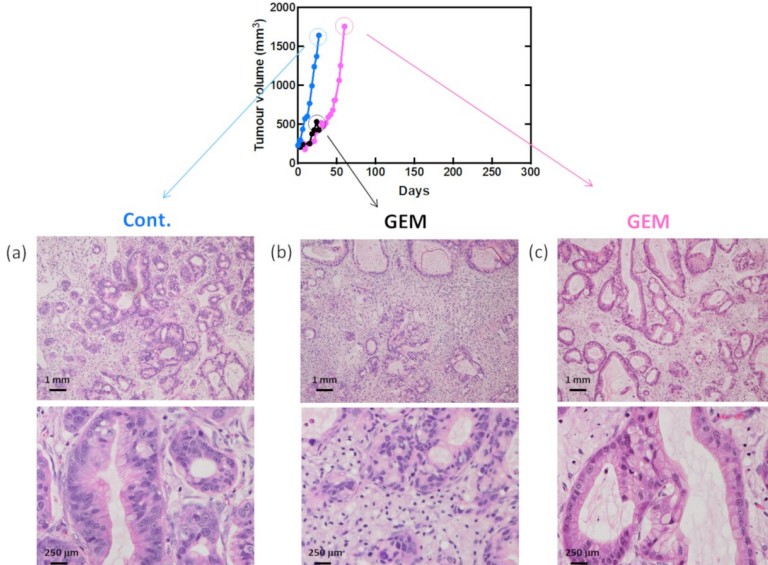

**Fig 3. Verification of antitumor effects in PDX tumors.** Pathological findings at each point after GEM treatment. Tumor growth curves after GEM treatment in PDX mice (#1). (a) Control tumor, (b) GEM-treated tumor. The tumor was grown for the same duration as the control. (c) GEM-treated tumor. The tumor was grown until reaching the same size as the control tumor. (a–c) Upper photographs are low magnification, and lower photographs are high magnification.

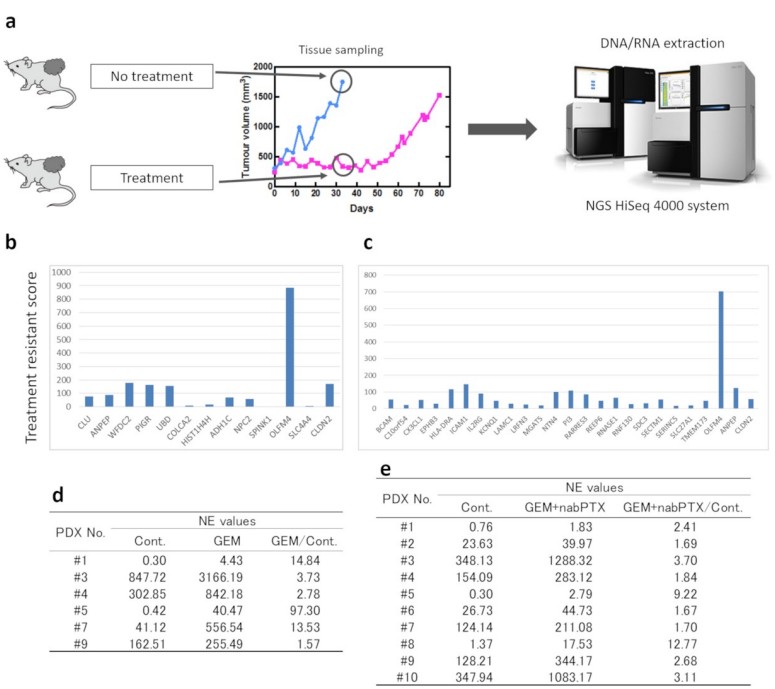

**Fig 4. Identification of chemotherapy resistance molecules.** (a) Schematic of the procedure for NGS analysis. (b, c) NGS analysis for the GEM administration and GEM + nab-PTX administration groups. Treatment resistance score was defined as the NE value ratio (treated group / untreated group) × NE value difference (treated group–untreated group). (d, e) The NE value of *OLFM4* mRNA. The ratio of NE values for treated and control groups were greater than 1.0 for all lines of PDXs. GEM, gemcitabine. nab-PTX, nab-paclitaxel.

nab-PTX combination therapy (10 lines) were analyzed, and genes with NE values of more than 10 and NE ratios (treated group to untreated group) of greater than 2 were selected (Fig 4B and 4C). From this analysis, we identified *OLFM4* as being strongly expressed in the treatment group.

The NE values for *OLFM4* are shown in Fig 4D and 4E. The ratio of NE values for *OLFM4* expression between the treated and control groups was greater than 1.0 for all PDX lines treated with GEM or GEM plus nab-PTX. Therefore, these findings demonstrated that the mRNA expression of *OLFM4* was higher in the treated group than in the control group, suggesting that *OLFM4* was highly expressed in chemoresistant tumors in the PDX model.

## Prognostic analysis from TCGA RNA database for pancreatic cancer

We also conducted additional studies using TCGA database. In total, 176 pancreatic cancer samples from TCGA database were evaluated for analysis of OLFM4 expression and OS. Fig 5 shows patient survival according to *OLFM4* mRNA expression. Patients were divided into the high and low *OLFM4* mRNA expression groups, and the prognoses of each group were examined. Patients in the OLFM4 low expression group exhibited significantly better survival rates than patients in the OLFM4 high expression group ($p$ = 0.0478).

## IHC staining for OLFM4 in the PDX model

Although we identified a chemoresistant molecule, *OLFM4*, at the mRNA levels using NGS analysis, we next analyzed the expression of OLFM4 protein using IHC (Fig 6A and 6B). GEM-treated PDXs showed higher expression of OLFM4 than untreated PDXs, similar to the results of *OLFM4* expression at the mRNA level.

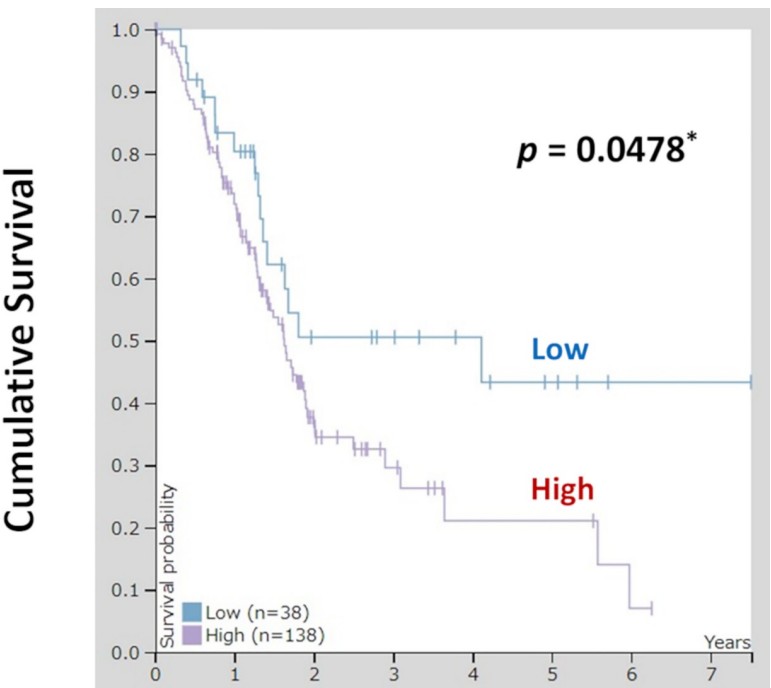

**Fig 5. Kaplan-Meier plots summarizing the results from analysis of the correlations between *OLFM4* mRNA expression and patient survival in TCGA pancreatic cancer database (n = 176).** Red line: high expression (n = 138), blue line: low expression (n = 38).

### In vitro experiment using tumor cells

To evaluate the role of OLFM4 in chemoresistance, we identified several cell lines in which endogenous OLFM4 gene expression was low and high for further transient overexpression and siRNA-mediated knockdown experiments, respectively (Fig 7A–7D). We measured the level of OLFM4 gene expression in HeLa cells and four pancreatic cancer cell lines (PANC-1, KP2, MIA PaCa-2, and SUIT-2) by real-time RT-PCR and found that endogenous OLFM4 expression was relatively lower in PANC-1 and MIA PaCa-2 cells than in KP2 and SUIT-2 cells (S4A Fig). Then, we conducted cell viability assay in response to GEM treatment. We selected HeLa cells and MIA PaCa-2 for the experiments of inducing exogenous OLFM4 expression, and SUIT-2 for the experiments of siRNA-mediated knockdown of OLFM4 depending on their transfection efficiency (S4B Fig, S4C Fig). Intriguingly, presence of exogenous OLFM4 in HeLa and MIA Paca-2 cells significantly increased cell viability in response to GEM compared to that in controls (Fig 7B and 7C). In contrast, cell viability was statistically lower in OLFM4-knockdown SUIT-2 cells via siRNA than in control cells (Fig 7D). These results indicated that OLFM4 might have an essential role in chemoresistance to GEM treatment in cancer cells including pancreatic cancer.

### Immunohistological analysis and prognostic analysis in patients with pancreatic cancer

Although we used the PDX model to study pancreatic cancer in this report, we also analyzed whether the expression of OLFM4 was observed at the protein level in specimens from patients

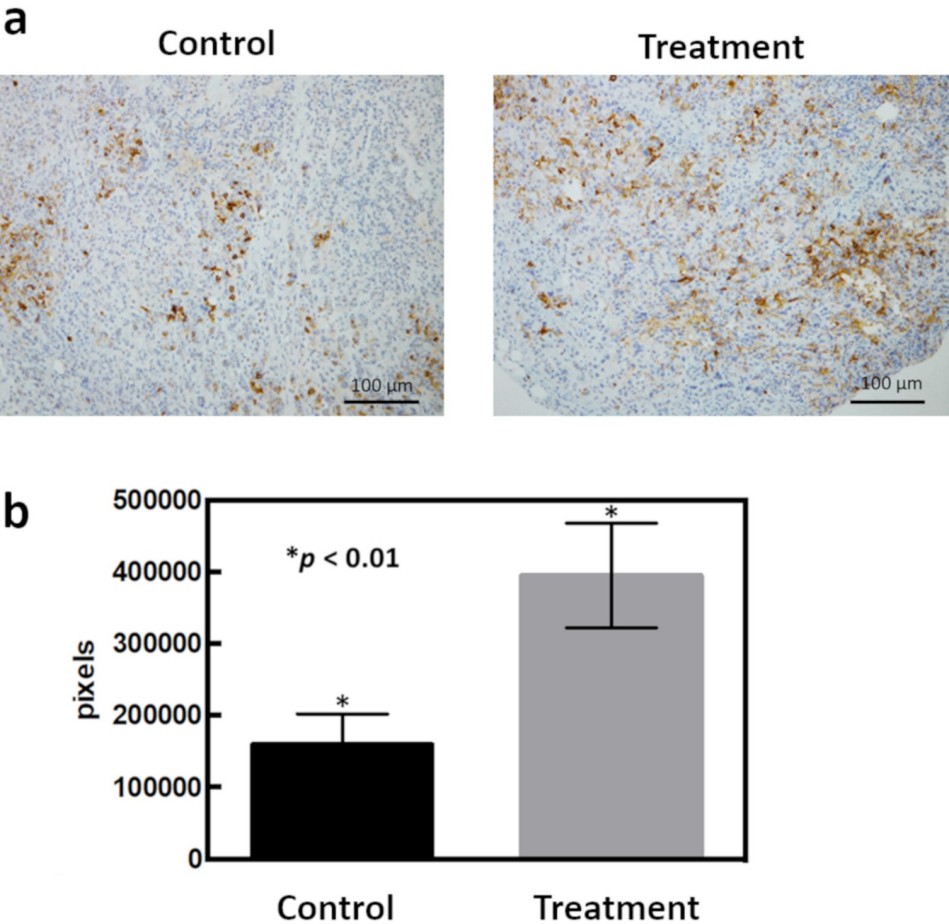

**Fig 6. Strong OLFM4 immunostaining was detected in chemotherapy-administered PDXs.** (a) Immunostaining for OLFM4 in PDXs. Control and chemotherapy-administered PDXs are shown at 200× each. (b) Analysis of the number of pixels of OLFM4-stained cells using Image J.

with pancreatic cancer. The clinicopathological features of the patients are summarized in Table 2. As shown in Fig 8A, OLFM4 staining was observed at the cell membrane and in the cytoplasm of tumor cells, and OLFM4 was also expressed in some stromal cells and normal glandular tissues. The samples were then divided into the high expression group (65.0% [52/80]) and low expression group (35.0% [28/80]).

Furthermore, we then analyzed whether OLFM4 was useful for the prognostic evaluation of pancreatic cancer. Notably, there were no significant relationships between high and low OLFM4 expression groups, including age ($p$ = 0.46), sex ($p$ = 0.061), tumor location (head or body/tail, $p$ = 0.37), histological type (adenocarcinoma or others, $p$ = 0.52), TNM stage (IIA or IIB, $p$ = 1.00), lymphatic invasion ($p$ = 0.68), venous invasion ($p$ = 0.11), and presence of adjuvant chemotherapy ($p$ = 0.94).

To investigate the prognostic value of OLFM4 expression in pancreatic cancer, we assessed the associations between OLFM4 expression levels and patient survival using Kaplan-Meier analysis with log-rank tests. As shown in Fig 8C, in the 80 patients with pancreatic cancer, those with OLFM4 low expression had better survival rates than patients with OLFM4 high expression ($p$ = 0.0296). Univariate and multivariate analyses are shown in Table 3. Univariate analysis of OS identified three prognostic parameters: sex ($p$ = 0.033), adjuvant chemotherapy ($p$ = 0.036), and OLFM4 expression ($p$ = 0.035), whereas multivariate analysis using Cox

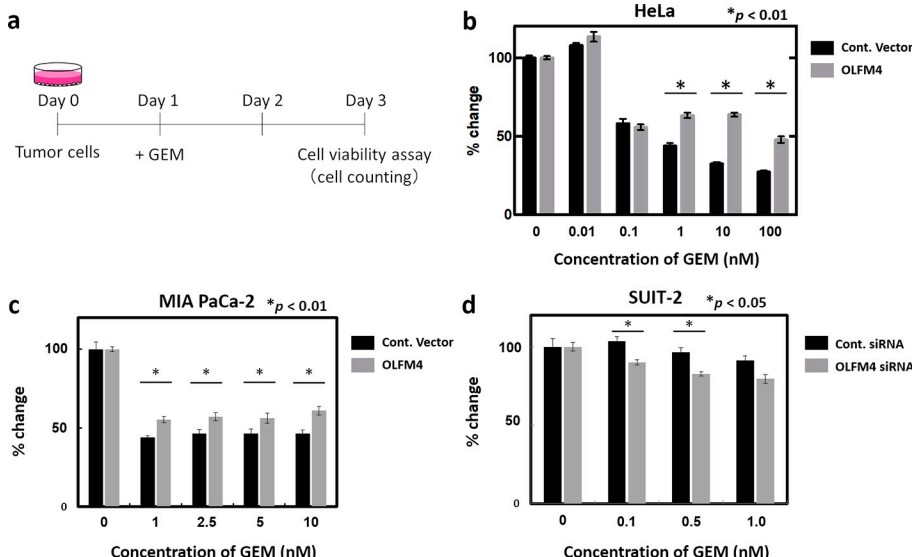

**Fig 7. Cell viability assay using cancer cell lines.** (a) Schematic representation of the procedure. Expression of the control vector and OLFM4 was induced in indicated cell lines. After 24 h (day 1), GEM was added at various concentrations. Cell viability assays were performed 48 h after GEM administration (day 3). (b and c) Rate of change of each measured OD value of the control vector and OLFM4-expressing HeLa cells (b) and MIA Paca2 cells (c) is shown. (d) Rate of change of each measured OD value of siRNA negative control and specific siRNA targeting OLFM4 induced in SUIT-2 cells is shown.

proportional hazards showed that adjuvant chemotherapy ($p = 0.028$) and OLFM4 expression ($p = 0.044$) were independent prognostic factors of poor outcomes in patients with pancreatic cancer.

## Discussion

Pancreatic cancer is an aggressive malignancy often characterized by delayed diagnosis, early metastasis, and chemoresistance. Therefore, it is important to identify novel prognostic biomarkers and therapeutic targets. Accordingly, in this study, we first established pancreatic cancer PDXs and identified OLFM4 as a chemoresistance-related molecule in PDXs treated with anticancer drugs. We performed NGS of tumor tissues from PDXs after treatment with standard chemotherapy. For both the GEM and GEM + nab-PTX administration group, few secondary gene mutations resulted from chemotherapy administration, as expected (data not shown). Moreover, all of these were passenger mutations, and no driver mutations were detected. Furthermore, no common gene mutations were found between chemotherapy groups. Accordingly, we concluded that although chemoresistance was not due to genetic mutations after chemotherapy administration, expression of OLFM4 mRNA was affected. In addition, in vitro experiments in OLFM4-expressing HeLa cells treated with GEM showed that OLFM4 expression was correlated with chemoresistance. Furthermore, we conducted cell viability assay for GEM-treated pancreatic cancer cell lines with both endogenous OLFM4 downregulation and upregulation. It was confirmed that OLFM4 expression had a role in chemoresistance via an vitro experiment using pancreatic cancer cell lines. In experiments using real human samples, we showed that high expression of OLFM4 was a critical independent prognostic factor and was associated with cancer survival in pancreatic cancer. Analysis of TCGA data also supported our results, highlighting OLFM4 expression as a prognostic factor in pancreatic cancer. Overall, these data showed that OLFM4 was a chemoresistance-related

**Table 2. Correlation between OLFM4 expression and clinicopathological features in 80 cases of pancreatic cancer.**

| Characteristics | OLFM4 expression | | P value |
|---|---|---|---|
| | **Low** **n = 28** | **High** **n = 52** | |
| Age (years) (mean ± SD) | 72.00 ± 9.38 | 71.75 ± 12.49 | 0.46[a] |
| Sex | | | 0.0608[b] |
| Male | 10 | 30 | |
| Female | 18 | 22 | |
| Tumor Location | | | 0.367[b] |
| Head | 16 | 35 | |
| Body/Tail | 12 | 17 | |
| Histological type | | | 0.5187[b] |
| Adenocarcinoma | 26 | 50 | |
| Others | 2 | 2 | |
| TNM (UICC 7th) | | | 1.00[b] |
| IIA | 7 | 13 | |
| IIB | 21 | 39 | |
| Lymphatic invasion | | | 0.6826[b] |
| ly0, ly1 | 10 | 21 | |
| ly2, ly3 | 18 | 31 | |
| Venous invasion | | | 0.1107[b] |
| v0, v1 | 1 | 8 | |
| v2, v3 | 27 | 44 | |
| Adjuvant chemotherapy | | | 0.9425[b] |
| Absent | 17 | 32 | |
| Present | 11 | 20 | |

[a]Student's t-test.

[b]Pearson chi-square test.

molecule and that OLFM4 expression was correlated with prognosis in patients with pancreatic cancer.

Various mouse tumor models and tumor cell lines have been used to study cancer. However, the results of these studies do not necessarily reflect human clinical data [21], mainly because of the lack of a tumor microenvironment in such cell and tumor models. Moreover, studies in mice often are not directly applicable to human patients [22,23]. Therefore, it is necessary to establish animal models that better reflect human clinical pathology. Recently, the NCI retired their "NCI-60" panel of human tumor cell lines that had been used for more than 25 years as an anticancer drug-screening platform. Instead, they have refocused drug screening on newer models, including PDXs, which are generated by implanting small chunks of human tumors in mice to create an environment that better mimics the human body [24]. Our group established 61 PDX lines from 116 surgically removed tumor tissues inoculated subcutaneously into NSG mice at the Kanagawa Cancer Center [17]. Therefore, PDX establishment success rate was 53% in our group. For pancreatic cancer, 10 PDX lines were established from 19 surgically removed pancreatic cancer tumor tissues, with a success rate of 52.6%. In this study, we established pancreatic cancer PDXs that preserved the pathological and genetic characteristics of human tumor tissues by transplanting patient-derived tumor specimens into super-immunodeficient mice. We confirmed that the PDXs retained most of the main histological and genetic characteristics of their donor tumors and remained stable after repeated passaging.

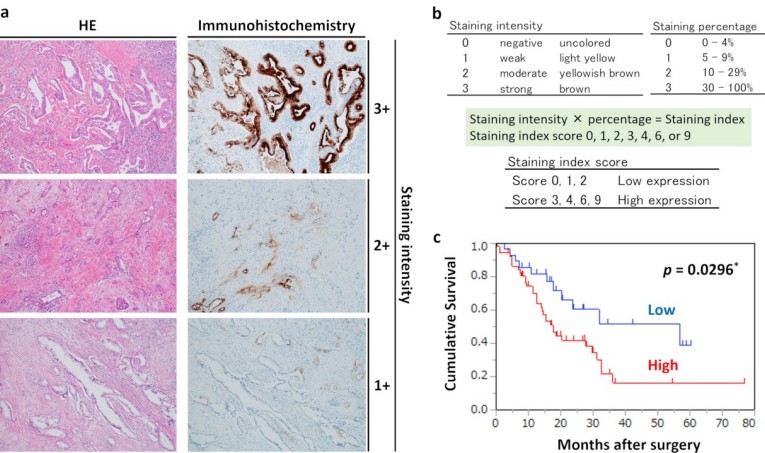

**Fig 8. Relationship between OLFM4 expression and prognosis.** (a) Immunohistochemical staining for OLFM4 in pancreatic cancer tissues (magnification: 100×). The left and right figures are the same sample tissue blocks and correspond to staining intensity. Left: HE staining. Right: immunohistochemical staining for OLFM4. (b) Criteria for determination of OLFM4 expression levels. OLFM4 expression levels for immunostaining were determined based on the intensity of staining and percentage of stained cells. Staining intensity and staining percentage criteria are shown. (c) Kaplan-Meier survival analysis in patients with pancreatic cancer (n = 80), showing overall survival according to OLFM4 protein expression. Red line: high expression group (n = 52); blue line: low expression group (n = 28).

Furthermore, we demonstrated that tumor cells of PDXs were derived from human patient tissues, not mice, by IHC analysis of the HLA class I molecule. Particularly for pancreatic cancer, it is often difficult to perform surgical resection and biopsy using endoscopic ultrasound/fine needle aspiration at the time of diagnosis; thus, the amount of sample tissue that can be collected is often limited. Therefore, PDX models are considered optimal animal models for research on pancreatic cancer.

**Table 3. Univariate and multivariate analyses of prognostic factor for overall survival in 80 patients with pancreatic cancer.**

| Clinicopathological factors | Univariate analysis | | | Multivariate analysis | | |
|---|---|---|---|---|---|---|
| | HR | 95% CI | *P* value[a] | HR | 95% CI | *P* value[a] |
| Age (≤ 71 versus > 71) (years) | 1.1 | 0.59–2.04 | 0.76 | - | - | - |
| Gender (male versus female) | 0.52 | 0.28–0.95 | 0.033[b] | 0.56 | 0.31–1.04 | 0.068 |
| TNM stage UICC 7th (IIA versus IIB) | 0.55 | 0.26–1.16 | 0.12 | - | - | - |
| Tumor location (body/tail versus head) | 1.57 | 0.83–2.95 | 0.16 | - | - | - |
| Lymphatic invasion (ly0, ly1 versus ly2, ly3) | 1.4 | 0.76–2.56 | 0.28 | - | - | - |
| Venous invasion (v0, v1 versus v2, v3) | 1.13 | 0.47–2.69 | 0.78 | - | - | - |
| Adjuvant chemotherapy (absent versus present) | 0.49 | 0.25–0.95 | 0.036[b] | 0.47 | 0.24–0.92 | 0.028[b] |
| OLFM4 (low versus high) | 2.1 | 1.05–4.19 | 0.035[b] | 2.06 | 1.02–4.15 | 0.044[b] |

95% CI, 95% confidence interval.

[a]Cox proportional hazard model.

[b]Statistically significant.

Using these PDXs, we found that increased mRNA expression of *OLFM4* led to chemoresistance. However, variations in mRNA expression do not always correspond to changes in protein expression because protein expression can be influenced by various post-transcriptional regulatory mechanisms [25–27]. Thus, mRNA measurement can be a poor predictor of protein abundance variations. Zhang et al. analyzed proteomics of colon and rectal tumors previously characterized in the TCGA database and showed that mRNA transcript abundance did not reliably predict protein abundance differences between tumors [28]. That is, mRNA expression levels do not necessarily correlate with protein expression or disease progression [29].

Next, we performed IHC staining for OLFM4 protein using PDX models to confirm whether OLFM4 protein levels were similar to mRNA levels. OLFM4 protein expression was higher in anticancer drug-treated PDXs than in untreated PDXs. Thus, these findings confirmed that OLFM4 was strongly expressed in anticancer drug-treated PDXs at the protein and mRNA levels.

Although PDX models show pathological features similar to those of the original human tumor, these models still do not completely reflect human pathology. Therefore, we examined the expression of OLFM4 protein and the relationships between OLFM4 expression and clinicopathological factors using 80 pancreatic cancer tissues from human patients. Clinical stage greatly affected prognosis in patients with pancreatic cancer; therefore, we focused on patients with stage IIA or IIB disease, which are the most frequently encountered patients as candidates for curative resection [30,31]. Currently, GEM monotherapy and GEM and nab-PTX combination therapy are not approved as neoadjuvant chemotherapies in Japan. Therefore, chemotherapy naïve patients were eligible for this study. Importantly, high expression of OLFM4 was an independent prognostic factor for survival in patients with pancreatic cancer.

Several studies have investigated the role of OLFM4 in tumor differentiation [32]. Studies based on the differentiation state of gastric and cervical cancers have demonstrated that an increase in the expression of OLFM4 is correlated with tumor differentiation state. These studies, carried out using IHC analysis for OLFM4 expression, showed a marked increase in OLFM4 in well-differentiated tissues, whereas OLFM4 levels were significantly lower in poorly differentiated tissues [33, 34]. In our study, OLFM4 expression did not differ depending on the degree of tumor differentiation. Therefore, OLFM4 expression may differ between pancreatic cancer and gastric or cervical cancer.

OLFM4 expression has been reported in several solid cancers and has been shown to be involved in prognosis [35–45]. However, high OLFM4 expression has been shown to be related to both poor and good prognoses in different studies, and no consensus has been reached. For instance, Mayama et al. showed that OLFM4 independently predicted a poor prognosis for ER-positive breast cancer. On the other hand, some studies have suggested that OLFM4-positive gastric and colorectal cancer patients have better survival rates than OLFM4-negative patients [35, 37, 46]. Therefore, the correlation between OLFM4 expression and cancer prognosis is controversial and has not yet been confirmed. In this study, we demonstrated that high expression of OLFM4 was significantly associated with poor prognosis in pancreatic cancer.

The contribution of OLFM4 to poor prognosis in pancreatic cancer may be related to the function of OLFM4. OLFM4 has been shown to participate in regulating cellular functions, such as cell proliferation [47–49] and differentiation [50]. In the PANC-1 cell line, OLFM4 mRNA was shown to increase through the early S phase of the cell cycle, and to promote proliferation by supporting the S to G2/M phase transition [47]. Moreover, OLFM4 also interacts with cell surface proteins, such as cadherin and lectins, and is involved in cell adhesion and migration [48, 51]. Furthermore, OLFM4 has been shown to inhibit apoptosis-promoting factor GRIM-19 to induce anti-apoptosis [52] and anti-apoptotic effects in tumor cells exposed to

stress-inducing factors, such as hydrogen peroxide, tumor necrosis factor-α, and cytotoxic agents [52–55]. These functions suggest that OLFM4 is involved in poor prognosis in pancreatic cancer and supported the results in our study.

In recent studies for patients, Yan et al. reported that OLFM4 was found to be significantly over-expressed in peripheral blood mononuclear cells (PBMCs) in pancreatic cancer patients, compared with a healthy control group [56]. They concluded that OLFM4 expression in peripheral blood could be a promising tumor marker for early detection of pancreatic cancer. In this study, we performed IHC analysis of OLFM4 expression using 80 pancreatic cancer tissues, instead of peripheral blood samples. In addition, we demonstrated that expression of OLFM4 was predictive of chemoresistance and a poor prognostic biomarker.

In addition, Takadate et al. identified four proteins, including OLFM4, as candidate prognostic markers of postoperative pancreatic ductal adenocarcinoma (PDAC) using a mass spectrometry-based proteomics approach with archived FFPE tissues. They found that high OLFM4 protein expression was correlated with significantly worse overall survival [39]. Although our study supported this result, the process for identifying candidate molecules that may have a poor prognosis was different. Using NGS, we found that the expression of *OLFM4* mRNA was upregulated after administration of anticancer drugs. Additionally, tumor cells expressing OLFM4 are resistant to anticancer drugs *in vitro* and may show variations in sensitivity to chemotherapy. Thus, it is possible that such cells expressing OLFM4 may persist after anticancer drug treatment, contributing to the observed poor prognosis. We demonstrated the role of OLFM4 in chemoresistance, which has not been described in previous studies. Thus, to determine the relationship between OLFM4 expression and chemoresistance, further cellular and molecular biological investigations are required. However, overall our findings indicate that OLFM4 may be a promising new prognostic marker and therapeutic target for pancreatic cancer.

## Supporting information

**S1 Table. Sources of tumor tissues from patients with pancreatic cancer.**
(XLS)

**S1 Fig. Establishment of patient-derived xenografts.**
(PPTX)

**S2 Fig. Verification of antitumor effects in PDX tumors.**
(PPTX)

**S3 Fig. Histopathological analysis of primary tumors and PDX tumors.**
(PPTX)

**S4 Fig. Quantitative reverse-transcription PCR and immunoblotting.**
(PPTX)

**S1 Text. Material & methods for S4 Fig.**
(DOC)

## Acknowledgments

This study is supported by research funds from Noile-Immune Biotech Inc.

## Author Contributions

**Conceptualization:** Satoshi Wada.

**Data curation:** Ryotaro Ohkuma, Erica Yada, Shumpei Ishikawa, Daisuke Komura, Kiyohiro Ando, Tetsuro Sasada, Tomoko Norose, Nobuyuki Ohike, Masafumi Takimoto, Shinichi Kobayashi, Satoshi Wada.

**Formal analysis:** Ryotaro Ohkuma, Erica Yada, Shumpei Ishikawa, Daisuke Komura, Yutaro Kubota, Kazuyuki Hamada, Hiroo Ishida, Yuya Hirasawa, Hirotsugu Ariizumi, Etsuko Satoh, Midori Shida, Makoto Watanabe, Rie Onoue, Kiyohiro Ando, Junji Tsurutani, Kiyoshi Yoshimura, Takehiko Yokobori, Tetsuro Sasada, Tomoko Norose, Nobuyuki Ohike, Masafumi Takimoto, Shinichi Kobayashi, Takuya Tsunoda, Satoshi Wada.

**Funding acquisition:** Satoshi Wada.

**Investigation:** Ryotaro Ohkuma, Erica Yada, Koji Tamada, Yutaro Kubota, Kazuyuki Hamada, Hiroo Ishida, Yuya Hirasawa, Hirotsugu Ariizumi, Etsuko Satoh, Tetsuro Sasada, Takeshi Aoki, Masahiko Murakami, Tomoko Norose, Nobuyuki Ohike, Masafumi Takimoto, Takuya Tsunoda, Satoshi Wada.

**Methodology:** Ryotaro Ohkuma, Erica Yada, Shumpei Ishikawa, Daisuke Komura, Hidenobu Ishizaki, Yutaro Kubota, Kazuyuki Hamada, Hiroo Ishida, Yuya Hirasawa, Hirotsugu Ariizumi, Etsuko Satoh, Midori Shida, Makoto Watanabe, Rie Onoue, Kiyohiro Ando, Junji Tsurutani, Kiyoshi Yoshimura, Tetsuro Sasada, Tomoko Norose, Nobuyuki Ohike, Masafumi Takimoto, Shinichi Kobayashi, Takuya Tsunoda, Satoshi Wada.

**Project administration:** Hidenobu Ishizaki, Koji Tamada, Satoshi Wada.

**Resources:** Yutaro Kubota, Kazuyuki Hamada, Hiroo Ishida, Yuya Hirasawa, Hirotsugu Ariizumi, Etsuko Satoh, Takeshi Aoki, Masahiko Murakami, Tomoko Norose, Nobuyuki Ohike, Masafumi Takimoto, Takuya Tsunoda, Satoshi Wada.

**Supervision:** Masahiko Izumizaki, Takuya Tsunoda, Satoshi Wada.

**Validation:** Takehiko Yokobori, Satoshi Wada.

**Writing – original draft:** Ryotaro Ohkuma.

**Writing – review & editing:** Ryotaro Ohkuma, Takuya Tsunoda, Satoshi Wada.

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
