## [Decision Letter · Decision Letter 0]

3 Oct 2019

PONE-D-19-18437

High expression of olfactomedin-4 is correlated with chemoresistance and poor prognosis in pancreatic cancer

PLOS ONE

Dear Prof. Wada,

Thank you for submitting your manuscript to PLOS ONE. After careful consideration, we feel that it has merit but does not fully meet PLOS ONE’s publication criteria as it currently stands. Therefore, we invite you to submit a revised version of the manuscript that addresses the points raised during the review process.

It looks that your manuscript was of little interest to the scientific community, because 27 scholars, including Dr. Gulley, declined or were reluctant to evaluate your report, except Reviewer 1.  Therefore, this academic editor evaluated your report by himself as another Reviewer.

You need to address the following issues:

(1) Next-generation sequencing (NGS).  You need to carry out whole-exome sequencing and analyze genetic alterations to investigate the mechanisms of chemoresistance in pancreatic cancer patients.

(2) OLFM4 overexpression experiments in HeLa cells.  You should describe how you established HeLa cells overexpressing OLFM4.  

(3) OLFM4 overexpression and knockdown experiments in pancreatic cancer cells.  You need to investigate OLFM4 expression levels in pancreatic cancer cell lines, and then carry out OLFM4 overexpression experiments in pancreatic cancer cell lines with endogenous OLFM4 downregulation, and also OLFM4 knockdown experiments in pancreatic cancer cell lines with endogenous OLFM4 upregulation.

(4) Pancreatic cancer PDXs.  You are advised to describe success rate of PDX establishment in your group.

(5) Histochemical analyses.  You showed histochemical analysis of primary tumor and PDXs in one case.  You need to show results for other cases in a supplementary Figure.

We would appreciate receiving your revised manuscript by Nov 17 2019 11:59PM. To enhance the reproducibility of your results, we recommend that if applicable you deposit your laboratory protocols in protocols.io, where a protocol can be assigned its own identifier (DOI) such that it can be cited independently in the future. For instructions see: http://journals.plos.org/plosone/s/submission-guidelines#loc-laboratory-protocols

We look forward to receiving your revised manuscript.

Kind regards,

Masaru Katoh, M.D., Ph.D.

Academic Editor

PLOS ONE

**Journal Requirements:**

2. We note that one or more of the authors are employed by a commercial company: Noile-Immune Biotech, Inc

3. We note that you are reporting an analysis of a microarray, next-generation sequencing, or deep sequencing data set. PLOS requires that authors comply with field-specific standards for preparation, recording, and deposition of data in repositories appropriate to their field. Please upload these data to a stable, public repository (such as ArrayExpress, Gene Expression Omnibus (GEO), DNA Data Bank of Japan (DDBJ), NCBI GenBank, NCBI Sequence Read Archive, or EMBL Nucleotide Sequence Database (ENA)). In your revised cover letter, please provide the relevant accession numbers that may be used to access these data. For a full list of recommended repositories, see http://journals.plos.org/plosone/s/data-availability#loc-omics or http://journals.plos.org/plosone/s/data-availability#loc-sequencing.

**Comments to the Author**

1. Is the manuscript technically sound, and do the data support the conclusions?

Reviewer #1: Yes

2. Has the statistical analysis been performed appropriately and rigorously? 

Reviewer #1: Yes

3. Have the authors made all data underlying the findings in their manuscript fully available?

Reviewer #1: Yes

4. Is the manuscript presented in an intelligible fashion and written in standard English?

Reviewer #1: Yes

5. Review Comments to the Author

Reviewer #1: Ohkuma and colleagues reported that high expression of OLFM4 was involved in chemoresistance and was an independent prognostic factor in pancreatic cancer. The study is interesting and the manuscript is well-written. My only concern is that the discussion is mostly covered by the repetition of the results and should be improved. The authors should explain the similarities and differences between the current study and published literature.

-Wang XY, et al. World J Gastroenterol. 2018;24(17):1881-1887.

-Guette C, et al. Proteomics Clin Appl. 2015;9(1-2):58-63.

-Yan H, et al. Hepatogastroenterology. 2011;58(109):1354-9.

6. PLOS authors have the option to publish the peer review history of their article (what does this mean?). If published, this will include your full peer review and any attached files.

Reviewer #1: No

---

## [Author Response · Author response to Decision Letter 0]

25 Nov 2019

Response to the Editor’s and Reviewer’s Comments

Editor:

(1) Next-generation sequencing (NGS). You need to carry out whole-exome sequencing and analyze genetic alterations to investigate the mechanisms of chemoresistance in pancreatic cancer patients.

Answer: 

We sincerely appreciate your comment.

We performed next-generation sequencing (NGS) of tumor tissues from PDXs before and after treatment with standard chemotherapy. Regarding Figure 2, in the Gemcitabine administration group, the number of gene mutations (missense mutations) after chemotherapy administration compared to before treatment was 1 (PDX #1), 1 (#3), 1 (#4), 11 (#5), 2 (#7) and 2 (#9). For the Gemcitabine + nab-Paclitaxel administration group, the number of gene mutations after chemotherapy administration was 4 (#1), 7 (#2), 1 (#3), 0 (#4), 2 (#5), 1 (#6), 1 (#7), 6 (#8), 1 (#9), and 0 (#10). Therefore, the number of secondary gene mutations was less than expected due to the administration of anticancer drugs. All secondary gene mutations were of the passenger type, and no driver mutations were detected. Furthermore, no common gene mutations were found between chemotherapy groups. From the above, we concluded that although chemoresistance was not due to genetic mutations after chemotherapy administration, OLFM4 mRNA expression was affected. We appreciate your helpful comment, and have added an explanation to the discussion section (P27-28, lines 452-457).

(2) OLFM4 overexpression experiments in HeLa cells. You should describe how you established HeLa cells overexpressing OLFM4. 

Answer:

Thank you for your helpful comment. HeLa cells were transfected with plasmid containing DNA of interest using FuGENE 6 Transfection Reagent (Progema, Madison, WI, USA), following the manufacturer’s protocol. For the OLFM4 transfection assay, HeLa cells were seeded in a 6-well Falcon plate. After 24 hours, cells were transfected with pCMV6-Ac-OLFM4-GFP tag plasmids (NM_006418, OriGene Technologies, Inc., Rockville, MD, USA) and an empty vector as a control, according to the manufacturer’s protocol. As indicated, we have included an explanation for how to establish HeLa cells overexpressing OLFM4 in the Material and Methods section (P13, lines 204-211).

(3) OLFM4 overexpression and knockdown experiments in pancreatic cancer cells. You need to investigate OLFM4 expression levels in pancreatic cancer cell lines, and then carry out OLFM4 overexpression experiments in pancreatic cancer cell lines with endogenous OLFM4 downregulation, and also OLFM4 knockdown experiments in pancreatic cancer cell lines with endogenous OLFM4 upregulation.

Answer:

　We sincerely appreciate your comment. According to your helpful comment, we carried out the additional experiments for using pancreatic cancer cell lines.

We investigated the assay of OLFM4 expression levels in pancreatic cancer cell lines, such as PANC-1, KP2, MIA PaCa-2 and SUIT-2 cell line. In addition, we demonstrated the cell viability assay for Gemcitabine addition in PANC-1 and MIA PaCa-2 cell line with endogenous OLFM4 downregulation, and in KP2 and SUIT-2 cell line with endogenous OLFM4 upregulation.

PANC-1 and MIA PaCa-2 cells were transfected pCMV-6-OLFM4-GFP and an empty vector, then it was confirmed OLFM4 overexpressed in only MIA PaCa-2 cells. Then we performed the cell viability assay for MIA PaCa-2 cells, the OLFM4 overexpressing cells were more remained than the control cells after adding Gemcitabine. Additionally, KP2 and SUIT-2 cells were transfected siRNA targeting OLFM4. The OLFM4 expression was knockdown in SUIT-2 cell lines, but OLFM4 knockdown was not observed in KP2 cell lines. We performed the cell viability assay for SUIT-2 cells, the OLFM4 knockdown cells were less viable than the control cells after adding Gemcitabine. From these results, we demonstrated the chemoresistant role for OLFM4 expression even in pancreatic cancer cell lines both with endogenous OLFM4 downregulation and upregulation.

As indicated, we have changed the Figure 7b-d (Fig 7) and the Figure Legends (P23, lines 393-399). We have included an explanation for the additional experiment in the section of Material and Methods (P12-14, lines 193-221) and Results (P22-23, lines 377-391). Supplementary Figure 4 (S4 Fig) was newly created to explain the results of the additional experiments. And based on the results of additional experiments, we have modified the part of Discussion section (P28, lines 459-462).

(4) Pancreatic cancer PDXs. You are advised to describe success rate of PDX establishment in your group.

Answer:

Thank you for your helpful comment. We established 61 PDX lines from 116 surgically removed tumor tissues inoculated subcutaneously into NSG mice at the Kanagawa Cancer Center [Chijiwa T, et al. Int J Oncol. 2015;47: 61-70]. The success rate was 53%. Regarding PDX for pancreatic cancer, 10 PDX lines were established from 19 surgically removed pancreatic cancer tumor tissues, and the success rate was 52.6%. As indicated, we have added an explanation for the PDX establishment success rate in the Discussion section (P29, lines 477-482).

(5) Histochemical analyses. You showed histochemical analysis of primary tumor and PDXs in one case. You need to show results for other cases in a supplementary Figure.

Answer:

　Thank you for your helpful comment. Since HE staining was performed in all cases, we presented them as a supplementary Figure 3 (S3 Fig). In nine cases other than Figure 1a, the histopathological characteristics of the patient tumor tissues matched that of PDX models. We appreciate your helpful comment, and have added an explanation to the section of Results (P17, lines 291-292).

Reviewer #1: 

Ohkuma and colleagues reported that high expression of OLFM4 was involved in chemoresistance and was an independent prognostic factor in pancreatic cancer. The study is interesting and the manuscript is well-written. My only concern is that the discussion is mostly covered by the repetition of the results and should be improved. The authors should explain the similarities and differences between the current study and published literature.

-Wang XY, et al. World J Gastroenterol. 2018;24(17):1881-1887.

-Guette C, et al. Proteomics Clin Appl. 2015;9(1-2):58-63.

-Yan H, et al. Hepatogastroenterology. 2011;58(109):1354-9. 

Answer:

We sincerely appreciate your helpful comment. We partially modified the discussion section to explain the similarities and differences between recent studies and our study.

　 Several studies have investigated the role of OLFM4 in tumor differentiation [Guette C, et al. Proteomics Clin Appl. 2015;9(1-2):58-6]. Increased expression of OLFM4 has been found to correlate with tumor differentiation state in gastric and cervical cancers, as determined through IHC analysis. In our study, the expression of OLFM4 did not differ according to degree of tumor differentiation.

　OLFM4 expression has been reported in several solid cancers and has been shown to be predictive of prognosis. One study found that OLFM4 independently predicted a poor prognosis in ER-positive breast cancer [Mayama et al. Cancer Sci. 2018;109: 3350-3359]. On the other hand, OLFM4 is correlated with a better prognosis in gastric and colorectal cancers [Wang XY, et al. World J Gastroenterol. 2018;24(17):1881-1887]. Therefore, the correlation between OLFM4 expression and cancer prognosis is controversial and has not yet been confirmed. In our study, we demonstrated that high expression of OLFM4 was significantly associated with poor prognosis in pancreatic cancer through IHC analysis.

　Yan H et al. demonstrated that OLFM4 was significantly over-expressed in peripheral blood mononuclear cells (PBMCs) in pancreatic cancer patients, compared with a healthy control group [Yan H, et al. Hepatogastroenterology. 2011;58(109):1354-9]. In this study, we performed IHC analysis of OLFM4 expression using 80 pancreatic cancer tissues instead of peripheral blood samples. We then demonstrated a correlation between expression of OLFM4 and chemoresistance/poor prognosis in pancreatic cancer.

　We really appreciate this helpful comment. As indicated, we have modified the Discussion section (P31-33, lines 517-534, 537-539, 542-552, 562-567).

---

## [Decision Letter · Decision Letter 1]

5 Dec 2019

High expression of olfactomedin-4 is correlated with chemoresistance and poor prognosis in pancreatic cancer

PONE-D-19-18437R1

Dear Dr. Wada,

We are pleased to inform you that your manuscript has been judged scientifically suitable for publication and will be formally accepted for publication once it complies with all outstanding technical requirements.

With kind regards,

Masaru Katoh, M.D., Ph.D.

Academic Editor

PLOS ONE

Additional Editor Comments (optional):

Reviewers' comments:

Reviewer's Responses to Questions

**Comments to the Author**

1. If the authors have adequately addressed your comments raised in a previous round of review and you feel that this manuscript is now acceptable for publication, you may indicate that here to bypass the “Comments to the Author” section, enter your conflict of interest statement in the “Confidential to Editor” section, and submit your "Accept" recommendation.

Reviewer #1: All comments have been addressed

2. Is the manuscript technically sound, and do the data support the conclusions?

Reviewer #1: Yes

3. Has the statistical analysis been performed appropriately and rigorously? 

Reviewer #1: Yes

4. Have the authors made all data underlying the findings in their manuscript fully available?

Reviewer #1: Yes

5. Is the manuscript presented in an intelligible fashion and written in standard English?

Reviewer #1: Yes

6. Review Comments to the Author

Reviewer #1: The authors have refined the relevant experiments and discussion, and the manuscript is acceptable for publication.

7. PLOS authors have the option to publish the peer review history of their article (what does this mean?). If published, this will include your full peer review and any attached files.

Reviewer #1: No

---

## [Editor Report · Acceptance letter]

18 Dec 2019

PONE-D-19-18437R1 

High expression of olfactomedin-4 is correlated with chemoresistance and poor prognosis in pancreatic cancer 

Dear Dr. Wada:

I am pleased to inform you that your manuscript has been deemed suitable for publication in PLOS ONE. Congratulations! Your manuscript is now with our production department. 

With kind regards,

on behalf of

Dr. Masaru Katoh 

Academic Editor

PLOS ONE